# Inflammatory Markers as Predictors of Shunt Dependency and Functional Outcome in Patients with Aneurysmal Subarachnoid Hemorrhage

**DOI:** 10.3390/biomedicines11040997

**Published:** 2023-03-23

**Authors:** Nina Rostgaard, Markus Harboe Olsen, Tenna Capion, Nanna MacAulay, Marianne Juhler

**Affiliations:** 1Department of Neurosurgery, The Neuroscience Centre Copenhagen University Hospital—Rigshospitalet, 2100 Copenhagen, Denmark; nina.rostgaard@regionh.dk (N.R.); tenna.baek.capion@regionh.dk (T.C.); 2Department of Neuroanaesthesiology, The Neuroscience Centre Copenhagen University Hospital—Rigshospitalet, 2100 Copenhagen, Denmark; markus.harboe.olsen@regionh.dk; 3Department of Neuroscience, University of Copenhagen, 2200 Copenhagen, Denmark; macaulay@sund.ku.dk

**Keywords:** cerebrospinal fluid, biomarkers, neuroinflammation, subarachnoid hemorrhage, post-hemorrhagic hydrocephalus

## Abstract

The mechanisms underlying post-hemorrhagic hydrocephalus (PHH) development following subarachnoid hemorrhage (SAH) are not fully understood, which complicates informed clinical decisions regarding the duration of external ventricular drain (EVD) treatment and prevents the prediction of shunt-dependency in the individual patient. The aim of this study was to identify potential inflammatory cerebrospinal fluid (CSF) biomarkers of PHH and, thus, shunt-dependency and functional outcome in patients with SAH. This study was a prospective observational study designed to evaluate inflammatory markers in ventricular CSF. In total, 31 Patients with SAH who required an EVD between June 2019 and September 2021 at the Department of Neurosurgery, Rigshospitalet, Copenhagen, Denmark, were included. CSF samples were collected twice from each patient and analyzed for 92 inflammatory markers via proximity extension assay (PEA), and the prognostic ability of the markers was investigated. In total, 12 patients developed PHH, while 19 were weaned from their EVD. Their 6-month functional outcome was determined with the modified Rankin Scale. Of the 92 analyzed inflammatory biomarkers, 79 were identified in the samples. Seven markers (SCF, OPG, LAP TGFβ1, Flt3L, FGF19, CST5, and CSF1) were found to be predictors of shunt dependency, and four markers (TNFα, CXCL5, CCL20, and IL8) were found to be predictors of functional outcome. In this study, we identified promising inflammatory biomarkers that are able to predict (i) the functional outcome in patients with SAH and (ii) the development of PHH and, thus, the shunt dependency of the individual patients. These inflammatory markers may have the potential to be employed as predictive biomarkers of shunt dependency and functional outcome following SAH and could, as such, be applied in the clinic.

## 1. Introduction

Non-traumatic subarachnoid hemorrhage (SAH) due to aneurism rupture is associated with high mortality and morbidity [1]. Despite recent advances in aneurysm treatment and neurocritical care, more than one third of SAH patients still develop an unfavorable long-term functional outcome [2]. Several complications contribute to the risk for an unfavorable outcome, including rebleeding, cerebral vasospasms, post-hemorrhagic hydrocephalus (PHH), seizures, delayed ischemic neurological deficits, cortical spreading depression, delayed cerebral ischemia, infections, cardiomyopathy, and pulmonary edema [2,3]. Shortly after the aneurysmal rupture, the intracranial pressure (ICP) rises [4,5], and blood extravasates into the cerebrospinal fluid (CSF) spaces, which can cause acute hydrocephalus with further elevation of ICP due to mechanical blockage to CSF circulation, CSF hypersecretion, or meningeal inflammation [6,7,8]. This condition can be treated with external ventricular drainage (EVD) [9,10]. The relief of CSF pressure through EVD is often necessary for many days, and it comes with a risk of bacterial central nervous system (CNS) infection through the percutaneous drain access; it has been shown that the risk of infection increases with the duration of EVD [11,12,13,14]. Intuitively, a CNS infection superimposed on the vascular event is likely to adversely affect the outcome, and the need for keeping the EVD for CSF pressure relief weighed against removal of the drainage as early as possible to reduce the infection risk is often a difficult balance in clinical practice. Accordingly, there is no consensus on recommendations for drainage duration or for EVD weaning via gradual increase in the drainage height vs. prompt closure of the EVD [9,10,15,16].

Although many of the SAH patients are successfully weaned from EVD due to the subsequent normalization of CSF flow dynamics, a considerable portion of these patients fail weaning from EVD and need to undergo shunt surgery due to development of PHH. This patient group is referred to as ‘shunt-dependent’. A recent review finds that the reported risk of shunt-dependency—i.e., PHH—after SAH varies between 8 and 63% [17]. This wide variation implies uncertainty about the prediction of the risk for chronic PHH requiring shunt insertion. Even though several publications have investigated clinical, radiological, and treatment features as predictive indicators for PHH following SAH, no biomarkers are available to actually predict shunt dependency [18,19,20,21,22,23].

Immediately after acute brain injury and hemorrhages such as SAH, local and systemic inflammatory responses trigger inflammatory signaling cascades accompanied by the activation and infiltration of immune cells of the brain, microglia, and astrocytes at the site of injury, as well as damage-associated pattern molecules (DAMPs) [24]. The induction of inflammatory cascades upon intracerebral hemorrhage may contribute significantly to the development of PHH, both via promotion of CSF hypersecretion in the CSF-secreting tissue in the ventricles, the choroid plexus, and via impaired reabsorption of the CSF as a result of scarring and obstruction of CSF drainage pathways [25]. In recent years, CSF biomarkers in SAH have come into interest, as several studies have found an increase in inflammatory markers in patients developing PHH secondary to SAH and other types of intracerebral hemorrhages [8,26,27], although these findings were often based on the analysis of only a small subset of inflammatory markers in their cohorts. A recent systematic review of such studies illustrated an increase in the inflammatory markers interleukin 6 (IL6), interleukin 18 (IL18), and vascular endothelial growth factor (VEGF) in patients with PHH compared to control subjects [26]. We subsequently employed a multiplex analysis to demonstrate a range of elevated inflammatory markers in the CSF from patients with SAH compared to control subjects undergoing clipping of non-ruptured aneurisms [8]. However, to our knowledge, whether CSF levels of inflammatory markers can predict the shunt dependency or outcome of patients suffering from SAH remains unresolved. [25,26].

In order to minimize the duration of EVD treatment and to find a clinical tool to support the prediction of shunt-dependency and/or the functional outcome of the patients, we aimed to identify potential inflammatory biomarkers in CSF samples that were obtained from patients with SAH in the acute phase of the disease and again before EVD removal or shunt insertion by analyzing the samples with a panel of 92 inflammation-related proteins.

## 2. Materials and Methods

### 2.1. Patients and Sample Collection

This study was a prospective observational study that was designed to evaluate inflammatory markers in CSF. Patients with SAH who required an EVD between June 2019 and September 2021 at the Department of Neurosurgery, The Neuroscience Centre, Copenhagen University Hospital—Rigshospitalet, Copenhagen, Denmark, were sought and included. Oral and written informed consent were obtained from all patients or next of kin depending on the capacity of the patients. The study was approved by the ethics committee of the Capital Region of Denmark (H-19001474, 22 March 2019) and the Danish Data Protection Agency (VD-2019-210, 8 April 2019). Other analyses from a proportion of the samples have previously been reported [8,28]. CSF was collected from 31 patients (median age: 60 y; range: 27–77 y; F/M: 21/10) through the EVD into a sterile collection tube. The first sample (‘start sample’, approximately 5 mL CSF) was obtained in the acute phase, either within 24 h of ictus (*n* = 22) or as soon as possible thereafter (*n* = 9). The last sample (‘end sample’, 1–2 mL CSF) was obtained just before removal of the EVD, and days between the two samples were registered (average time 17.9 days, range 5–30 days). Upon collection, the CSF samples were centrifuged at 2000× *g* for 10 min immediately after sampling and aliquoted into polypropylene microtubes (Sarstedt, Nürnbrecht, Germany) before storage at −80 °C [29]. Nineteen patients did not develop chronic hydrocephalus and were weaned off EVD (‘weaned’ group), and twelve patients underwent ventriculoperitoneal shunt surgery due to PHH (‘shunt’ group). The functional outcome as classified by the modified Rankin Scale (mRS) was assessed for each patient six months after discharge from the neurosurgical department. Twelve patients had a favorable 6-month functional outcome (mRS 0–2), and twelve patients had an unfavorable functional outcome (mRS 3–6). Seven patients were lost to follow-up.

### 2.2. Inflammatory Panel

The CSF samples were analyzed for 92 inflammatory markers via a proximity extension assay (PEA; Olink Bioscience, Uppsala, Sweden, https://www.bioxpedia.com/wp-content/uploads/2020/04/1029-v1.3-Inflammation-Panel-Content.pdf, accessed on 15 March 2023) at BioXpedia A/S (Aarhus, Denmark) [30]. In brief, the PEA technique is a targeted protein screening and employs a pair of oligonucleotide-conjugated antibodies to detect each of the tested markers in a CSF volume of 1 µL. Each of the oligonucleotide antibody-pairs contains unique DNA sequences which allow hybridization only to each other. Upon detection, the oligonucleotides are brought into close proximity, hybridize, and enable DNA polymerization, which produces a PCR sequence. The PCR sequence is then amplified and quantified via real-time qPCR. Ct values from the qPCR are translated into a relative quantification unit, Normalized Protein eXpression (NPX) via computation. The NPX values are an arbitrary unit on a log_2_ scale. The limit of detection (LOD), which is the lowest measurable level of an individual inflammatory marker, was defined by three times the standard deviation of each marker over the background signal. The assay was performed in a blinded fashion regarding the patient groups and the study endpoint. Of the 92 inflammatory markers, 13 were not identified (below the LOD).

### 2.3. Statistical Analysis

Clinical characteristics were collected from the electronic patient chart and handled via REDCap version 12.0.33 (Research Electronic Data Capture, Vanderbilt University, Nashville, TN, USA). All data processing and statistical analyses were carried out by using R version 4.1.0 (R Core Team, Vienna, Austria). For each of the prediction models, patients were stratified into two groups, and only proteins that were detected in at least five patients of each group were included to obtain the predictive potential of each protein. The prognostic ability was investigated for the start sample, the end sample, and the average change between the start and end sample. Average change per day was calculated by identifying the change in the marker between those two samples for each participant and dividing it with the time between the start and end sample. The prognostic ability was investigated by using area under the curve (AUC) estimates that were calculated from a receiver operating characteristics (ROC) curve. The markers were labelled as acceptable predictors if the lower 95% confidence limit of the AUC was above 0.7 [31]. For those identified as acceptable predictors, the optimal cut-off was identified by using the Youden’s cut-off, which reflected the point where the combined sensitivity and specificity was the highest [32]. The cut-off, area under the curve, specificity, sensitivity, and corresponding 95% confidence intervals are presented for each of the inflammatory markers.

## 3. Results

The presence of inflammatory markers in the CSF obtained from SAH patients was detected with a proximity extension assay that consisted of a panel of 92 inflammatory markers [30], out of which 79 were identified in the patient CSF.

### 3.1. Shunt Dependency

To determine whether any of the identified inflammatory markers in the CSF obtained in either the first or the last sample could serve as a predictor of shunt dependency, we divided the patients into two groups: those who received a VP shunt (*n* = 12; F/M: 11/1; age 53–77 years) and those who were successfully weaned from EVD (*n* = 19; F/M: 10/9; age 27–74 years), and we compared the levels of quantified inflammatory markers between these two groups. In total, 57 inflammatory markers were detected in the *start* samples in at least 5 patients from both the VP-shunted group and the EVD group, 58 for the *end* samples, and 54 from both (Appendix A). Figure 1 illustrates AUC for the prediction of shunt dependency for each of the detected inflammatory markers. Overall, 7 of these reached a lower confidence limit ≥ 0.7 and could be considered valid predictors of shunt dependency (Table 1). For start samples, no markers were identified as valid predictors. For end samples, 6 markers were identified as valid predictors with AUCs (with 95% confidence interval (CI)) of 0.88 (0.77–1.00) for stem cell factor (SCF), 0.86 (0.74–0.99) for osteoprotegerin (OPG), 0.86 (0.72–0.99) for latency-associated peptide-transforming growth factor beta 1 (LAP TGFβ1), 0.86 (0.72–0.99) for FMS-related tyrosine kinase 3 ligand (Flt3L), 0.86 (0.73–1.00) for fibroblast growth factor 19 (FGF19), and 0.86 (0.72–0.99) for cystatin-D (CST5) (Table 1 and Figure 2). For the average change per day, (calculated from the change in the marker between the start and the end samples normalized to the number of days between the samples), only colony stimulating factor 1 (CSF1) was identified as a valid predictor with an AUC of 0.91 (0.79–1.00) (Table 1 and Figure 2).

### 3.2. Functional Outcome

To reveal potential biomarkers that could serve as predictors of the functional outcome following the hemorrhagic event, we quantified the inflammatory marker content of the start and end CSF samples and plotted these according to the functional outcome of the patient six months after ictus. In total, 7 of the 31 patients were lost to follow-up and were not included in this analysis. Again, only markers that were detected in at least five samples in both groups were included in the analysis. In total, 59 markers were detected in the start samples, 57 markers were detected in the end samples, and 54 markers showed up for both, thereby allowing computation of the average change per day (Appendix A). A total of 12 patients had a favorable functional outcome (mRS 0–2, see Section 2), and 12 had an unfavorable outcome (mRS 3–6). Figure 3 illustrates AUC for the prediction of the outcome for each of the detected inflammatory markers. Overall, 4 of these reached a lower confidence limit ≥ 0.7 and could be considered valid predictors of functional outcome following SAH (Table 2). For start samples, no markers were identified as valid predictors. For end samples, 3 markers were identified as valid predictors with AUCs (95% CI) of 0.86 (0.70–1.00) for tumor necrosis factor-alpha (TNFα), 0.91 (0.78–1) for C-X-C motif chemokine ligand 5 (CXCL5), and 0.92 (0.79–1.00) for chemokine C-C motif ligand 20 (CCL20) (Table 2 and Figure 4). For average change per day, only interleukin 8 (IL8) was identified as a valid predictor with an AUC of 0.86 (0.71–1.00) (Table 2 and Figure 4).

## 4. Discussion

### 4.1. Hydrocephalus and Shunt Dependency

The main finding of this study is that potential inflammatory biomarkers from an early point after the SAH bleed may predict the outcome related to shunt dependency and functional outcome. Although many of the SAH patients are successfully weaned from EVD due to the subsequent normalization of CSF flow dynamics, a considerable proportion of these patients fail weaning from EVD and need to undergo shunt surgery. Several publications have investigated clinical, radiological, and treatment features as predictive indicators for shunt-dependent hydrocephalus following SAH [18,19,20,21,22,23]. A recent review lists a large number of potential risk factors including patient age and gender; Glasgow Coma and Hunt-Hess scores at ictus; radiological signs of acute hydrocephalus; radiological assessment of bleeding severity including the extent of blood in the ventricular system; need for EVD insertion; CSF drainage volume; EVD weaning; vasospasm and/or cerebral infarction; and occurrence of fever [17]. It has also been proposed that surgical vs. endovascular aneurysm ligation could influence the risk for PHH, but a recent meta-analysis from 2019 disputes this [33]. Moreover, increased levels in the CSF of S100B [22] protein, erythrocyte count, and interleukin [34] might also predict the risk for shunt dependency after SAH. In recent years, neuroinflammation upon SAH as a contributing factor of PHH has come into focus, and a range of publications has demonstrated elevated levels of various inflammatory factors in CSF from patients with SAH compared to that obtained from healthy individuals [8,26,27]. Of these, IL1β, TNFα, and IL6/8 are key inflammatory markers that are often detected in the CSF from SAH patients [26,35,36,37,38,39] and may contribute to cerebral vasospasm [39]. Regrettably, IL1β is not included in the tested panel of inflammatory markers (https://www.bioxpedia.com/wp-content/uploads/2020/04/1029-v1.3-Inflammation-Panel-Content.pdf, accessed on 15 March 2023). Although we previously detected IL6 and IL8 (amongst other inflammatory markers) as elevated in CSF from SAH patients, TNFα was not significantly elevated following Bonferroni correction for multiple samples [8]. Nevertheless, none of these three biomarkers appears to serve as a predictor for shunt-dependency, and only IL8 and TNFα served as predictors for functional outcome in the present study. This finding contrasts that of an earlier study that indicated elevated IL6 levels in the early post-SAH period as a useful diagnostic tool for predicting shunt dependency in patients with acute PHH [40] and functional outcome [36].

In the present study, we found that the inflammatory markers SCF, OPG, LAP TGFβ1 Flt3L, FGF19, CST5, and CSF1 could be used as predictors of shunt dependency. In patients who received a shunt, the average levels were higher in the end samples for the majority of these markers (SCF, OPG, LAP TGFβ1 Flt3L, FGF19, CST5) compared to patients who could be weaned from CSF diversion. The average change in the levels of CSF1 from the start to the end sample differed between the patient group that developed PHH compared to the patient group that did not. This relationship suggests that the level of hemorrhage-related inflammation dictates the PHH formation and, thus, shunt dependency, which is in line with an earlier demonstration of the acute inflammatory responses occurring in intracerebral hemorrhage being a possible contributing factor of PHH [8,25,41].

### 4.2. Functional Outcome

Cerebral vasospasm and delayed cerebral ischemia (DCI) are major factors for survival and functional outcome following SAH and have, for many years, been the main focus in SAH research [42,43,44,45]. Clinical parameters such as Hunt and Hess scores, lower modified Fisher grade, the absence of intracerebral hematoma, intact pupillary light reflexes, and clinical improvement before aneurysm treatment can be used as early clinical predictors of functional outcome and favorable/non-favorable prognosis [46]. However, other factors such as inflammation have also been shown to be useful predictors of poor outcome in patients with SAH. Systemic inflammatory response syndrome score has been found to be an independent prognostic factor of poor functional outcome following SAH as assessed by mRs scores [47,48,49]. Of the inflammatory predictors, elevated CSF chemokine levels and inflammatory mediators showed an association with an unfavorable clinical outcome after SAH [35,50,51].

Here, we found that the inflammatory markers TNFα, CXCL5, CCL20, and IL8 were predictors of functional outcome. The first three markers were assessed as being at higher levels in the end sample of the group with an unfavorable functional outcome compared to those with a favorable functional outcome, and the last marker was assessed as having a larger average daily difference detected in the group with an unfavorable outcome.

### 4.3. Neuroinflammation and Therapeutic Targets

An imbalance of CSF drainage and CSF production may promote hydrocephalus. Emerging evidence suggests that neuroinflammation can mediate both reduced CSF drainage and increased CSF production [50]. The impaired CSF drainage can arise from fibrosis in the brain tissue, e.g., the arachnoid granulations in adults, thereby obstructing the outflow routes [52] and disordered arterial pulsatility may further contribute to a decreased CSF flow and, thus, modulate the fine-tuned CSF clearance apparatus [53]. In recent years, it has come into focus that neuroinflammation following a hemorrhagic event might promote hypersecretion of CSF, thus contributing to the development of PHH [25]. In rat models, it was proposed that following intraventricular hemorrhage (IVH), the inflammatory cascade resulted in hyperactivation of the ion (and fluid) transporters in the choroid plexus tissue and, hence, increased fluid transportation into the ventricles [8,41,54]. Upon intracerebral hemorrhage, the brain tissue is exposed to blood components. This results in the activation of microglia, which are the resident immune cells within the brain, and the recruitment of peripheral leukocytes (macrophages) [50]. Signals from the damaged brain region can lead to the activation of the systemic immune system, which is then followed by immunosuppression [50]. The early activation of the immune system may lead to secondary injury, resulting in further activation of microglia; secretion of proinflammatory cytokines, ROS, and matrix metalloproteinases; and neuronal injury [25,55], and a similar cascade seems probable following SAH. Cytokines are a class of small proteins that act as signaling molecules that regulate inflammation and modulate cellular activities such as growth, survival, and differentiation [56]. Cytokines do not only act to increase neuroinflammation upon brain hemorrhage but also contribute to PHH through fibrosis and scarring of the leptomeninges and arachnoid granulations, as well as protein deposition in the periventricular tissue [57,58].

The majority of the predictive markers found in this study belong to the cytokine family (TGFβ1, SCF, Flt3L, CSF1, TNFα), with one receptor of tumor necrosis factor (TNF) receptor superfamily (OPG). We detected a small number of chemokines (CXCL5, CCL20, and IL8), which are responsible for the induction of cell migration of immune cells upon inflammation [56]; a proteinase inhibitor controlling proteolytic activity during inflammation (CST5); and a growth factor involved in the processes of the adaptive and innate immune system (FGF-19). The binding of a cytokine or chemokine ligand to its receptor results in the activation of the receptor, which, in turn, triggers a cascade of signaling events that regulates a variety of cellular functions connected to immune response and inflammation [56].

Our finding that the predictive markers are mostly increased in the unfavorable outcome-groups (VP-shunt and mRS3–6) suggests that there is increased inflammation in these patients compared to the patients with a more favorable outcome. Hence, the inflammatory machinery could be a pharmacological therapeutic target to improve the outcome for patients following SAH. Several clinical trials with the immunosuppressant statin simvastatin have been conducted on patients with SAH to improve the outcome, although meta-analyses conducted on trial data found no beneficial effects of the treatment [59,60]. The statins and another immunosuppressant, cyclosporine, which also failed to prevent the unfavorable outcome following SAH [61], may not target the immune pathways involved in the acute inflammatory response following SAH. Thus, these immunosuppressants do not reverse the acute inflammatory cascade following bleeding and thereby fail to prevent the downstream negative effects of SAH resulting in unfavorable outcome for the patients.

The predictive markers found in this study do not only relate to neuroinflammation but may be present during systemic inflammation in general. Although these markers are known to be involved in inflammatory and/or immune responses, some of the molecules are implicated in other pathophysiological processes such as fibrosis in inflammatory conditions, i.e., liver, myocardial, and pulmonary fibrosis [62,63,64,65]. We cannot rule out that some of the markers are elevated as a result of fibrotic tissue formation around the EVD and, therefore, may vary with the number of days with EVD insertion. The varying number of days between the start and the end samples, which is indicative of days with EVD insertion (17.9 days, range 5–30), is therefore a limitation of this study. The sampling times were chosen for ethical reasons to prevent infection by puncturing the sterile drain system, thus only sampling when the patients underwent intervention as a part of their treatment. The increased risk of drain/CSF infection by repeated CSF sampling precluded daily sampling, which would have been optimal for a time profile in marker development. Instead, we have chosen to calculate an average per-day change in marker levels, but the change in marker levels may not be evenly spread over the study days. Since our CSF analysis was conducted, customized PEA panels including additional important neuroinflammatory markers (e.g., IL1β and IL1-RA) have become available and may provide further insight into neuroinflammation in connection with various neurological conditions [66].

## 5. Conclusions

In conclusion, we report a distinct inflammatory profile of the CSF obtained from SAH patients receiving a shunt and those with an unfavorable outcome. These inflammatory markers may have the potential to be employed as predictive biomarkers of shunt dependency and functional outcome following SAH and could, as such, be applied in the clinic. Future validation in a larger cohort is required to establish the relevance of the detected predictors in this study. In addition, increased insight into the neuroinflammatory pathways and downstream effectuators associated with SAH could reveal if a more specific treatment, aiming to decrease inflammation in SAH patients, could benefit patients in terms of better outcome and fewer cases of PHH. We here focused on inflammatory markers in the CSF, but elevated BBB permeability to immune cells could be equally important for shunt dependency and unfavorable outcome and a relevant target for future studies.

## Figures and Tables

**Figure 1 biomedicines-11-00997-f001:**
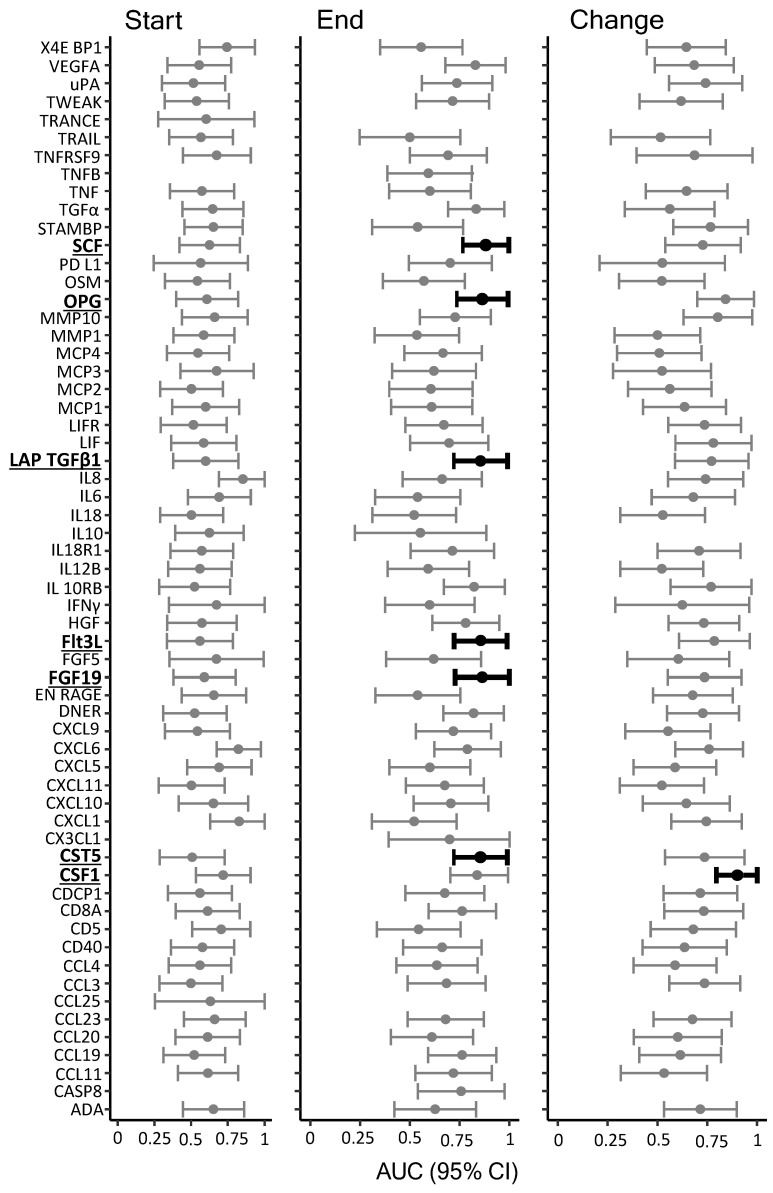
Inflammatory markers as predictors of shunt dependency. The figure illustrates AUC and 95% CI for prediction of shunt dependency for each of the detected inflammatory markers. The markers (highlighted/black bars) that reached the required level of ≥0.7 of the lower confidence limit were accepted as prognostic markers of development of hydrocephalus following SAH. EVD weaned: *n* = 19; VP-shunt. *n* = 12; AUC: area under the curve; CI: confidence interval.

**Figure 2 biomedicines-11-00997-f002:**
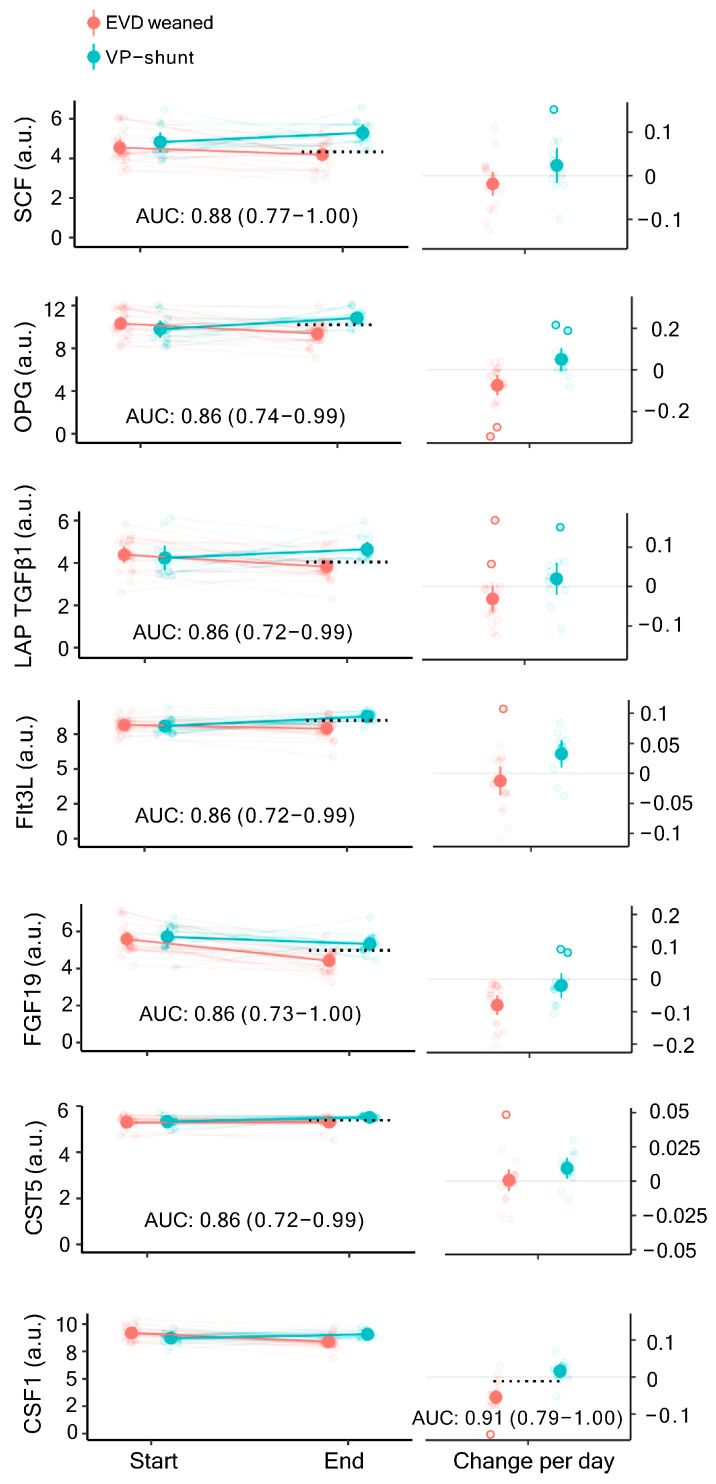
Valid predictors of shunt dependency. The abundance of the inflammatory markers in the start vs. end sample is shown in the left column, with AUCs (AUC: arbitrary units with 95% CI), and the daily change in abundance illustrated in the right column). Dashed lines show Youden’s threshold as a cutoff for sensitivity and specificity for the given marker. EVD weaned: *n* = 19; VP-shunt: *n* = 12; AUC: area under the curve; CI: confidence interval.

**Figure 3 biomedicines-11-00997-f003:**
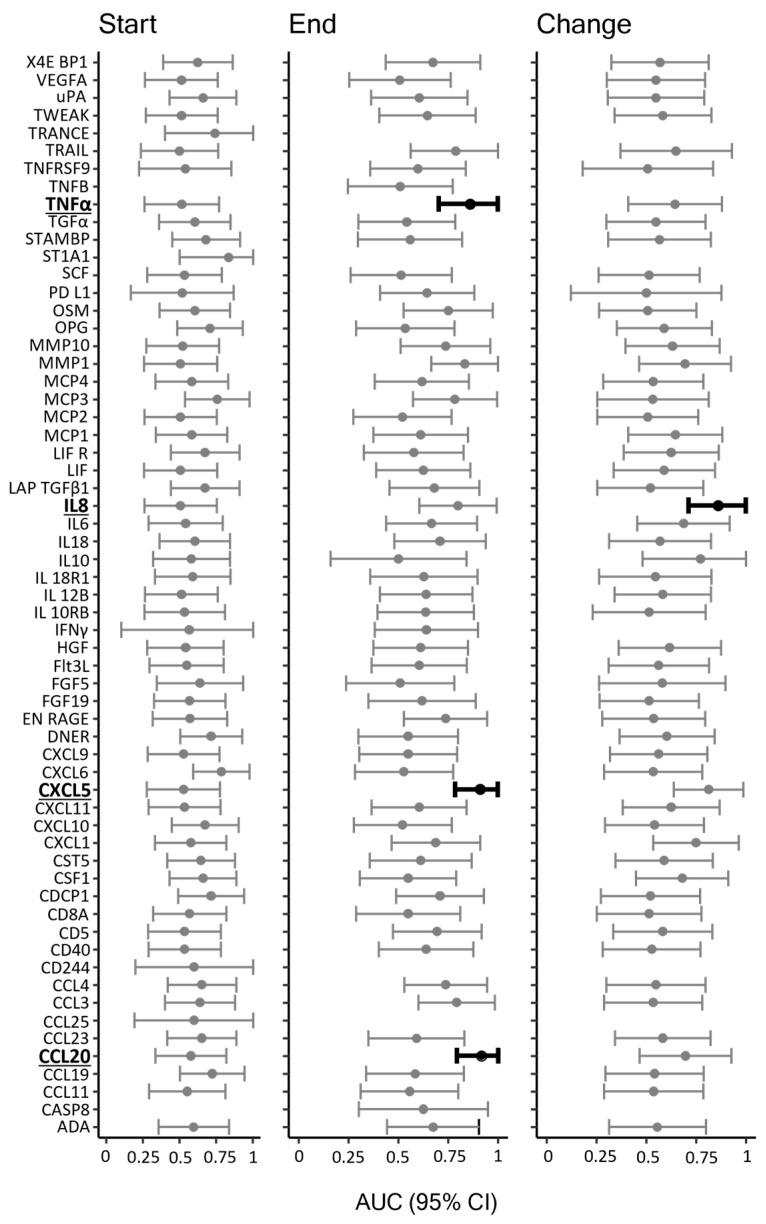
Inflammatory markers as predictors of functional outcome. The figure illustrates AUC and 95% CI for prediction of functional outcome for each of the detected inflammatory markers. The markers (highlighted/black bars) that reached the required level of ≥0.7 of the lower confidence limit were accepted as prognostic markers of functional outcome following SAH. Favorable functional outcome (mRS 0–2): *n* = 12; unfavorable functional outcome (mRS 3–6) *n* = 12; AUC: area under the curve; CI: confidence interval.

**Figure 4 biomedicines-11-00997-f004:**
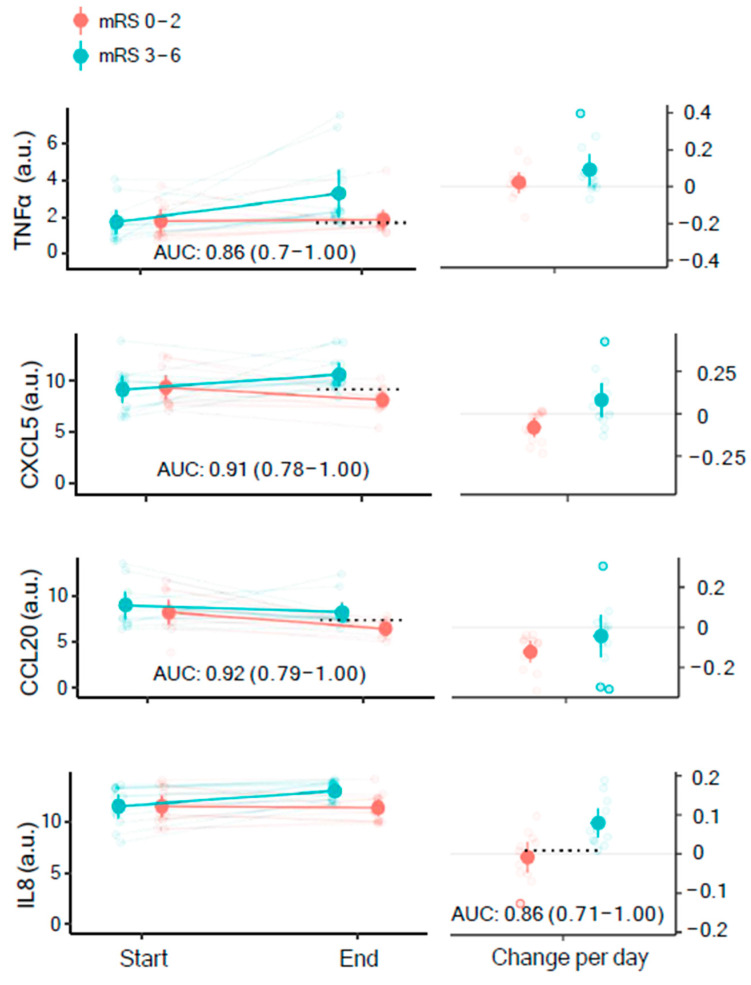
Valid predictors of functional outcome. The abundance of the inflammatory markers in the start vs. end sample is shown in the left column, with AUCs (AUC: arbitrary units with 95% CI), and the daily change in abundance illustrated in the right column). Dashed lines show Youden’s threshold as a cutoff for sensitivity and specificity for the given marker. Favorable functional outcome (mRS 0–2): *n* = 12; unfavorable functional outcome (mRS 3–6): *n* = 12; AUC: area under the curve; CI: confidence interval.

**Table 1 biomedicines-11-00997-t001:** Acceptable predictors of hydrocephalus measured by the Olink 96 inflammation panel. Inflammatory markers with AUC values above 0.7. AUC: area under the curve; CI: confidence interval; SD: standard deviation.

Name	Time	EVD Weaned Mean (SD) [*n*]	VP-Shunt Mean (SD) [*n*]	AUC (95% CI)	Cut-Off	Sensitivity	Specificity
SCF	End	4.18 (0.74) [19]	5.29 (0.65) [12]	0.88 (0.77–1.00)	4.33	1.00	0.63
OPG	End	9.37 (1.13) [19]	10.83 (0.80) [12]	0.86 (0.74–0.99)	10.20	0.83	0.79
LAP TGFβ1	End	3.83 (0.63) [19]	4.65 (0.57) [12]	0.86 (0.72–0.99)	4.05	1.00	0.68
Flt3L	End	7.90 (0.76) [19]	8.76 (0.37) [12]	0.86 (0.72–0.99)	8.49	0.83	0.79
FGF19	End	4.42 (0.63) [19]	5.31 (0.61) [12]	0.86 (0.73–1.00)	4.97	0.83	0.89
CST5	End	5.29 (0.25) [19]	5.51 (0.06) [12]	0.86 (0.72–0.99)	5.38	1.00	0.68
CSF1	Daily Change	0.05 (0.04) [19]	−0.02 (0.03) [12]	0.91 (0.79–1.00)	0.01	0.92	0.89

**Table 2 biomedicines-11-00997-t002:** Acceptable predictors of functional outcome measured by the Olink 96 inflammation panel. Inflammatory markers with AUC values above 0.7. AUC: area under the curve; CI: confidence interval; SD: standard deviation.

Name	Time	mRS 0–2 - Mean (SD) [*n*]	mRS 3–6 - Mean (SD) [*n*]	AUC (95% CI)	Cut-Off	Sensitivity	Specificity
TNFα	End	1.84 (0.88) [12]	3.26 (2.01) [12]	0.86 (0.7–1.00)	1.66	1.00	0.67
CXCL5	End	8.08 (1.23) [12]	10.56 (1.75) [12]	0.91 (0.78–1.00)	9.11	0.92	0.83
CCL20	End	6.33 (0.85) [11]	8.12 (1.69) [12]	0.92 (0.79–1.00)	7.26	0.92	0.91
IL8	Daily Change	0.01 (0.06) [12]	−0.08 (0.06) [12]	0.86 (0.71–1.00)	−0.01	1.00	0.58

## Data Availability

Anonymized data are available upon reasonable request to the corresponding author.

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
