# Peer review of "Inflammatory Markers as Predictors of Shunt Dependency and Functional Outcome in Patients with Aneurysmal Subarachnoid Hemorrhage"

_biomedicines, 2023, doi:10.3390/biomedicines11040997_

Round 1
Reviewer 1 Report
The study of Rostgaard et al is another biomarker paper about neuroinflamatory biomarkers post hemorragic stroke like the one that have been published recently, nowadays in hundreds already. Nevertheless the authors try to focus in finding predictors of success of ventricular drainage or the need of shunt, that by itself could be an interesting aim.
However the study suffers of certain issues that make difficult to interpret it's findings. The most relevant one is the absence of IL1beta on the panel, that have become the facto the gold standard for this type of biomarker study. The authors do not justify nowhere it's absence either.
Secondly, even taking into account the reduced sample numbers, the authors also fail to reproduce TNFalpha as indicator on the acute phase also validated everywhere. TNFalpha, that in figure 1 is only specified as "TNF".
Third, the markers that reach signification on the shunt/EVD part of the study are incidentally the ones with less dispersion, that accompanied with the low n make and the outliers on the significant predictors of figure 2, me think about outlier issues. Perhaps it will interesting to the authors to review their cohorts for concomitant diseases or treatments or simply age, and eventually remove some subjects.
Four, I've some methodological questions:
- did acute stroke biomarkers like CRP were similar for all patients?
- did the duration of the EVD was the same for all patients? how was incorporated into the statistics?
- in the methods it says "at least 5 samplings" but how these collection days where spaced? can you provide more details ?
- what happen to the 1/2 of the cohort that was lost? exitus? If the later,
why where not included in the followup as a third class of "unfavorable outcome"?
- the relation between unfavorable outcome and chronic inflammation pointed on the discussion is interesting, specifically regarding about BBB permeability to immune cells, and probably you have enough data to support it. But statistically you don't do it and you don't present it on the results.
Author Response
We thank the reviewers for their thorough evaluation of our manuscript. We have corrected the revised manuscript accordingly (marked in red font) and responded to the critique points below.
Reviewer 1
The study of Rostgaard et al is another biomarker paper about neuroinflamatory biomarkers post hemorragic stroke like the one that have been published recently, nowadays in hundreds already. Nevertheless the authors try to focus in finding predictors of success of ventricular drainage or the need of shunt, that by itself could be an interesting aim.
However the study suffers of certain issues that make difficult to interpret it's findings. The most relevant one is the absence of IL1beta on the panel, that have become the facto the gold standard for this type of biomarker study. The authors do not justify nowhere it's absence either.
Answer: We agree with the reviewer that IL1beta is an important neuroinflammatory agent and apologize for its absence. However, the panel of inflammatory markers that the samples were tested for simply does not contain that biomarker (https://www.bioxpedia.com/wp-content/uploads/2020/04/1029-v1.3-Inflammation-Panel-Content.pdf). We have now included a link to the tested panel (page 3, line 102) and included a statement regarding the IL1beta in SAH (page 10, line 232).
Secondly, even taking into account the reduced sample numbers, the authors also fail to reproduce TNFalpha as indicator on the acute phase also validated everywhere. TNFalpha, that in figure 1 is only specified as "TNF".
Answer: We agree that TNFalpha is indeed an important neuroinflammatory marker. In a previous study (Lolansen et al., FBCNS 2022), we compared inflammatory markers in the start samples (acute phase) from SAH patients compared to CSF from control individuals. TNF was indeed elevated in the SAH patients, if looked upon in singularity (p < 0.001, student’s t-test, Suppl. File 5). However, with a large panel of inflammatory markers, we performed Bonferroni correction on the statistics, and TNF was no longer significantly elevated under these conditions. We have included a statement to this effect in the revised Discussion, page 10, line 234.
Third, the markers that reach signification on the shunt/EVD part of the study are incidentally the ones with less dispersion, that accompanied with the low n make and the outliers on the significant predictors of figure 2, me think about outlier issues. Perhaps it will interesting to the authors to review their cohorts for concomitant diseases or treatments or simply age, and eventually remove some subjects.
Answer: We thank the reviewer for his/her suggestion and agree that it could be interesting with a more in depth analysis of the patients. However, with the low number of tested patients, we fear that our data would be less robust if we remove selected groups of patients.
Four, I've some methodological questions:
- did acute stroke biomarkers like CRP were similar for all patients?
Answer: We did not extract the baseline inflammatory parameters. The baseline distribution of inflammatory markers collected from blood samples and their prognostic values has been investigated previously (see for example Schuhmann et al J Neurosurg 2005, Lolak et al Surg Infect 2013). Our aim was to seek to identify new possible biomarkers of shunt dependency and functional outcome in CSF Fluid. We therefore hope omission of CRP is acceptable.
- did the duration of the EVD was the same for all patients? how was incorporated into the statistics?
Answer: The duration of the EVD varied between patients with an average time of 17.9 days between ‘start’ and ‘end’ sample (see Methods). The time between sampling has not been directly included in the statistics, but in Fig. 2 and 4, right panels, ‘change of inflammatory marker per day’ is related to the number of days between samples.
- in the methods it says "at least 5 samplings" but how these collection days where spaced? can you provide more details ?
Answer: We apologize for the lack of clarity of the text. We do not refer to five samplings, but state that we only include inflammatory markers that were detected in at least five patients of each group. We have rephrased this in the revised version of the manuscript, page 3, line 121:” For each of the prediction models, patients were stratified into two groups and only proteins that were detected in at least five patients of each group were included to obtain the predictive ability of each protein”.
- what happen to the 1/2 of the cohort that was lost? exitus? If the later,
why where not included in the followup as a third class of "unfavorable outcome"?
Answer: The patients were lost to follow-up, but not because of exitus and could therefore not be considered a combined group of ‘unfavorable outcome’. The number of patients lost to follow up is now stated in the revised method section, page 3, line 99.
- the relation between unfavorable outcome and chronic inflammation pointed on the discussion is interesting, specifically regarding about BBB permeability to immune cells, and probably you have enough data to support it. But statistically you don't do it and you don't present it on the results.
Answer: Another very interesting point that could be interesting to pursue for future studies. We do not feel confident, based on our present data, to conduct statistical analysis on this particular point and/or conclude on the extent of BBB permeability to immune cells in our patients. We have included this highly relevant consideration in the discussion, pages 11-12, line 333.
Reviewer 2 Report
This is a interesting study showing the inflammatory marker change in patients with SAH. The manuscript was well-written and the data were well-presented.
Author Response
We thank the reviewers for their thorough evaluation of our manuscript. We have corrected the revised manuscript accordingly (marked in red font) and responded to the critique points below.
Reviewer 2
This is a interesting study showing the inflammatory marker change in patients with SAH. The manuscript was well-written and the data were well-presented.
Answer: We thank the reviewer for his/her positive feedback.
Reviewer 3 Report
In this manuscript, the authors investigated different inflammatory biomarkers of shunt dependency and functional outcome by the analysis of cerebrospinal fluid (CSF) of posthemorrhagic hydrocephalus (PHH) in patients with subarachnoid hemorrhage (SAH). They found seven biomarkers (SCF, OPG, LAP TGFβ1, Flt3L, FGF19, CST5, and CSF1) as predictors of shunt dependency, whereas four markers (TNFα, CXCL5, CCL20 and IL8) as predictors of functional outcome. The authors mentioned that the identified inflammatory biomarkers could be clinically potential as predictors of shunt dependency and functional outcome in patients with SAH. The manuscript is well-written, and the interpretation of data is in good shape. The study is very interesting; however, the authors need to clarify the following minor issues:
1) It is better to briefly define the term “shunt dependency” in patients with subarachnoid hemorrhage (SAH) in the Introduction section for easy understand to the readers.
2) Please briefly provide the methodology of cerebrospinal fluid (CSF) collection with reference/s in the Materials and Methods section.
3) Please change “subarachnoidal hemorrhage” to “subarachnoid hemorrhage (page 1, lines 13 and 32), as well as change “The Non-traumatic” ---” to “The non-traumatic” (page 1, line 36).
4) Please mention the full form of “ICP” (page 2, line 45)
5) Please change “Fms related tyrosine kinase 3 ligand (Flt3L)” to “Fms-related tyrosine kinase 3 ligand (Flt3L)” (page32, lines 144-145). Also change “CSF-1”to “CSF1” (page 10, line 267).
6) As osteoprotegerin (OPG) is a member of the tumor necrosis factor (TNF) receptor superfamily, please clearly the meaning of the part of sentence “one receptor of cytokines (OPG)” (page 10, line 268).

Author Response
We thank the reviewers for their thorough evaluation of our manuscript. We have corrected the revised manuscript accordingly (marked in red font) and responded to the critique points below.
Reviewer 3
In this manuscript, the authors investigated different inflammatory biomarkers of shunt dependency and functional outcome by the analysis of cerebrospinal fluid (CSF) of posthemorrhagic hydrocephalus (PHH) in patients with subarachnoid hemorrhage (SAH). They found seven biomarkers (SCF, OPG, LAP TGFβ1, Flt3L, FGF19, CST5, and CSF1) as predictors of shunt dependency, whereas four markers (TNFα, CXCL5, CCL20 and IL8) as predictors of functional outcome. The authors mentioned that the identified inflammatory biomarkers could be clinically potential as predictors of shunt dependency and functional outcome in patients with SAH. The manuscript is well-written, and the interpretation of data is in good shape. The study is very interesting; however, the authors need to clarify the following minor issues:
1) It is better to briefly define the term “shunt dependency” in patients with subarachnoid hemorrhage (SAH) in the Introduction section for easy understand to the readers.
Answer: We thank the reviewer for his/her positive feedback. ‘Shunt dependency’ has now been defined in the revised introduction, p.2, line 59.
2) Please briefly provide the methodology of cerebrospinal fluid (CSF) collection with reference/s in the Materials and Methods section.
Answer: The CSF sampling methodology has been extended in the revised Methods section, page 2, line 86.
3) Please change “subarachnoidal hemorrhage” to “subarachnoid hemorrhage (page 1, lines 13 and 32), as well as change “The Non-traumatic” ---” to “The non-traumatic” (page 1, line 36).
Answer: We thank the reviewer for spotting the mistakes. These have been corrected in the revised version of the manuscript.
4) Please mention the full form of “ICP” (page 2, line 45)
Answer: We thank the reviewer for pointing this out and have included the abbreviation after first mention, page 1, line 43.
5) Please change “Fms related tyrosine kinase 3 ligand (Flt3L)” to “Fms-related tyrosine kinase 3 ligand (Flt3L)” (page32, lines 144-145). Also change “CSF-1”to “CSF1” (page 10, line 267).
Answer: We thank the reviewer for noticing these mistakes that have now been corrected in the revised version of the manuscript.
6) As osteoprotegerin (OPG) is a member of the tumor necrosis factor (TNF) receptor superfamily, please clearly the meaning of the part of sentence “one receptor of cytokines (OPG)” (page 10, line 268).
Answer: We thank the reviewer. The sentence has been clarified to correctly reflect OPG as a member of the TNF receptor superfamily, page 11, line 295.
Round 2
Reviewer 1 Report
Interesting choice made by the authors of one panel that have no reputation on the field, that does not include the analytes that everyone else is measuring and even more interestingly that the author's findings are precisely in analytes that aren't in the common neuroinflamation panels elsewhere (As Latency-associated peptide transforming growth
factor beta 1 (LAP TGF-beta-1) , Osteoprotegerin (OPG), Stem cell factor (SCF) and Fibroblast growth factor 19 (FGF-19)). May i ask why the authors decided for this obsolete panel? Last publication with it is already 4 years ago. Olink is nowadays customizing PEA panels that allow the detection of Il1-beta, Il1-RA, etc.. Check 10.1089/neur.2022.006 for example.
Furthermore that this set of analytes are not characteristic of neuroinflamation but instead fibrosis make the reviewer suspect that what you are detecting is simply the reactive fibrosis around your EVD catheter that somehow correlate with time the EVD has been in patient's brain -that also i asked you and you refuse to provide- and the EVD time it's very well known correlate of a positive outcome.
Author Response
We thank the reviewer for his/her thorough evaluation of our manuscript and the additional valuable points that are now included in the revised version of the manuscript (marked in red in the uploaded manuscript).
Reviewer 1, round 2:
Interesting choice made by the authors of one panel that have no reputation on the field, that does not include the analytes that everyone else is measuring and even more interestingly that the author's findings are precisely in analytes that aren't in the common neuroinflamation panels elsewhere (As Latency-associated peptide transforming growth
factor beta 1 (LAP TGF-beta-1) , Osteoprotegerin (OPG), Stem cell factor (SCF) and Fibroblast growth factor 19 (FGF-19)). May i ask why the authors decided for this obsolete panel? Last publication with it is already 4 years ago. Olink is nowadays customizing PEA panels that allow the detection of Il1-beta, Il1-RA, etc.. Check 10.1089/neur.2022.006 for example.
Furthermore that this set of analytes are not characteristic of neuroinflamation but instead fibrosis make the reviewer suspect that what you are detecting is simply the reactive fibrosis around your EVD catheter that somehow correlate with time the EVD has been in patient's brain -that also i asked you and you refuse to provide- and the EVD time it's very well known correlate of a positive outcome.
Answer: We are sorry to hear that the reviewer feels that way about the assay that we employed for the analysis. We chose a wide inflammation Olink PEA assay to detect as many inflammatory markers as possible. These are not specifically neuro-inflammation markers. We were not aware at the time that specific markers that could be of interest were missing from the panel. However, our study employed this analysis as a discovery tool to determine whether any of these 92 markers could be employed as a predictor for shunt dependency or outcome. That does not rule out that other markers may also serve such a role. We have included at statement to that effect in the Discussion of the revised manuscript, page 11.
Although the mentioned markers may well be associated with fibrosis, they nevertheless are known to be involved in inflammation as well, please see below. However, it is a very interesting point, and potential connection to the EVD that the reviewer brings up and a statement to that effect is now included in the revised Discussion, page 11.
We have absolutely no intensions of refusing to provide the time EVD in the patients’ brains. It has, all along, been stated in the introduction (average 17.9 days). We have now included the range as well (5-30 days), page 3 and introduced the varying amount of EVD time as a limitation to the study, pages 11. We thank the reviewer for this valuable point.
SCF (c-Kit-ligand): is a cytokine. Upregulated under inflammatory conditions. Has a role in release of histamine, pro-inflammatory cytokines and chemokines. Described in several inflammation-related diseases. Also implicated in fibrosis (In pulmonary fibrosis SCF-c-kit pathway was activated in BLM-injured lung and might play a direct role in pulmonary fibrosis by the recruitment of bone marrow progenitor cells capable of promoting lung myofibroblast differentiation. PMID 23401096), (cardiac fibrosis: Infusion of c-kit + resident cardiac stem cells markedly reduced myocardial fibrosis + reduced the rate of apoptosis and oxidative stress. PMID 26587804)
FGF-19: hormone and growth factor, involved in processes of the adaptive and innate immune system. Antifibrotic molecule (liver and pulmonary fibr.) with potential therapeutic interest in fibrotic lung disorders (PMID 35549849).
LAP TGF-beta-1 (proprotein for TGF-b1): multifunctional protein. Cytokine, performs many cellular functions (cell growth, cell proliferation, cell differentiation, and apoptosis). Following brain injury microglia and astrocytes produce large concentration of TGF-b1 (PMID 26867675). Also central mediator of fibrogenesis and upregulated and activated in fibrotic diseases. TGF-b correlated with poorer functional outcomes in elderly stroke patients + promotes basement membrane fibrosis (PMID 31721012).
Osteoprotegerin OPG: is a cytokine decoy-receptor. Belongs to TBF receptor super family. Up-regulated by estrogens and increasing calcium concentrations. Has a role in transcriptional regulation in inflammation, innate immunity and cell survival. Expressed in many tissues including dendritic cells of the immune system. (OPG and inflammation review, PMID 12101070). Implicated in organ fibrosis (liver, pulmonary, cardiac, renal). (OPG and fibrosis review, PMID 34171336)
